# Multimodal Attention for Layout Synthesis in Diverse Domains

## Abstract

We address the problem of scene layout generation for diverse domains such as images, mobile applications, documents and 3D objects. Most complex scenes, natural or human-designed, can be expressed as a meaningful arrangement of simpler compositional graphical primitives. Generating a new layout or extending an existing layout requires understanding the relationships between these primitives. To do this, we propose a multimodal attention framework, MMA, that leverages self-attention to learn contextual relationships between layout elements and generate novel layouts in a given domain. Our framework allows us to generate a new layout either from an empty set or from an initial seed set of primitives, and can easily scale to support an arbitrary of primitives per layout. Further, our analyses show that the model is able to automatically capture the semantic properties of the primitives. We propose simple improvements in both representation of layout primitives, as well as training methods to demonstrate competitive performance in very diverse data domains such as object bounding boxes in natural images (COCO bounding boxes), documents (PubLayNet), mobile applications (RICO dataset) as well as 3D shapes (PartNet).

## 1 Introduction

In the real world, there exists a strong relationship between different objects that are found in the same environment (Torralba & Sinha, 2001; Shrivastava & Gupta, 2016). For example, a dining table usually has chairs around it, a surfboard is found near the sea, horses do not ride cars, *etc.*. Biederman (2017) provided strong evidence in cognitive neuroscience that perceiving and understanding a scene involves two related processes: *perception* and *comprehension*. Perception deals with processing the visual signal or the appearance of a scene. Comprehension deals with understanding the *schema* of a scene, where this schema (or layout) can be characterized by contextual relationships (*e.g.*, support, occlusion, and relative likelihood, position, and size) between objects. For generative models that synthesize scenes, this evidence underpins the importance of two factors that contribute to the *realism* or plausibility of a generated scene: layout, *i.e.*, the arrangement of different objects, and their appearance (in terms of pixels). Therefore, generating a realistic scene necessitates both these factors to be plausible.

The advancements in the generative models for image synthesis have primarily targeted plausibility of the appearance signal by generating incredibly realistic images often with a single entity such as faces (Karras et al., 2019; 2017), or animals (Brock et al., 2018; Zhang et al., 2018). In the case of large and complex scenes, with a lot of strong non-local relationships between different elements, most methods require proxy representations for layouts to be provided as inputs (*e.g.*, scene graph, segmentation mask, sentence). We argue that to plausibly generate large and complex scenes without such proxies, it is necessary to understand and generate the layout of a scene, in terms of contextual relationships between various objects present in the scene.

The layout of a scene, capturing what primitives occupy what parts of the scene, is an incredibly rich representation. Learning to generate layouts itself is a challenging problem due to the variability of real-world or human-designed layouts. Each layout is composed of a small fraction of possible objects, objects can be present in a wide range of locations, the number of objects varies for each scene and so do the contextual relationships between objects.

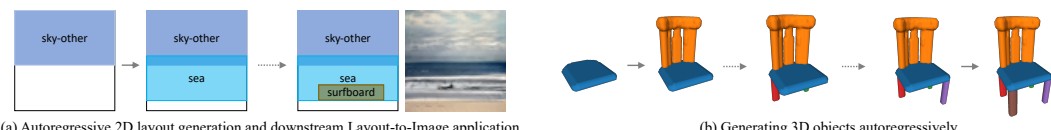

(a) Autoregressive 2D layout generation and downstream Layout-to-Image application          (b) Generating 3D objects autoregressively

Figure 1: Our framework can synthesize layouts in diverse natural as well as human designed data domains such as natural scenes or 3D objects in a sequential manner.

Formally, a scene layout can be represented as an unordered set of graphical primitives. The primitive itself can be discrete or continuous depending on the data domain. For example, in the case of layout of documents, primitives can be bounding boxes from discrete classes such as 'text', 'image', or 'caption', and in case of 3D objects, primitives can be 3D occupancy grids of parts of the object such as 'arm', 'leg', or 'back' in case of chairs. Additionally, in order to make the primitives compositional, we represent each primitive by a location vector with respect to the origin, and a scale vector that defines the bounding box enclosing the primitive. Again, based on the domain, these location and scale vectors can be 2D or 3D. A generative model for layouts should be able to look at all existing primitives and propose the placement and attributes of a new one. We propose a novel Multimodal Attention framework (MMA) that first maps the different parameters of the primitive independently to a fixed-length continuous latent vector, followed by a masked Transformer decoder to look at representations of existing primitives in layout and predict the next parameter. Our generative framework can start from an empty set, or a set of primitives, and can iteratively generate a new primitive one parameter at a time. Moreover, by predicting either to stop or to generate the next primitive, our sequential approach can generate variable length layouts.

Our approach can be readily plugged into scene generation frameworks (e.g., Layout2Image (Zhao et al., 2019), GauGAN (Park et al., 2019b)) or stand-alone applications that require generating layouts or templates with/without user interaction. For instance, in the UI design of mobile apps and websites, an automated model for generating plausible layouts can significantly decrease the manual effort and cost of building such apps and websites. Finally, a model to create layouts can potentially help generate synthetic data for various tasks tasks (Yang et al., 2017; Capobianco & Marinai, 2017; Chang et al., 2015; Wu et al., 2017b;a).

To the best of our knowledge, MMA is the first framework to perform competitively with the state-of-the-art approaches in 4 diverse data domains. We evaluate our model using existing metrics proposed for different domains such as Jensen-Shannon Divergence, Minimum matching distance, and Coverage in case of 3D objects, Inception Score and Fréchet Inception Distance for COCO, and Negative Log-likelihood of the test set in case of app wireframes and documents. Qualitative analysis of the framework also demonstrates that our model captures the semantic relationships between objects automatically (without explicitly using semantic embeddings like word2vec Mikolov et al. (2013)).

## 2   RELATED WORK

**Generative models.** Deep generative models based on CNNs such as variational auto-encoders (VAEs) (Kingma & Welling, 2013), and generative adversarial networks (GANs) (Goodfellow et al., 2014) have recently shown a great promise in terms of faithfully learning a given data distribution and sampling from it. There has also been research on generating data sequentially (Oord et al., 2016; Chen et al., 2020) even when the data has no natural order (Vinyals et al., 2015). Many of these approaches often rely on low-level information (Gupta et al., 2020b) such as pixels while generating images (Brock et al., 2018; Karras et al., 2019), videos (Vondrick et al., 2016), or 3D objects (Wu et al., 2016; Yang et al., 2019; Park et al., 2019a; Gupta et al., 2020a) and not on semantic and geometric structure in the data.

**Scene generation.** Generating 2D or 3D scenes conditioned on sentence (Li et al., 2019d; Zhang et al., 2017; Reed et al., 2016), a scene graph (Johnson et al., 2018; Li et al., 2019a; Ashual & Wolf, 2019), a layout (Dong et al., 2017; Hinz et al., 2019; Isola et al., 2016; Wang et al., 2018b) or an existing image (Lee et al., 2018) has drawn a great interest in vision community. Given the input, some works generate a fixed layout and diverse scenes (Zhao et al., 2019), while other works generate diverse layouts and scenes (Johnson et al., 2018; Li et al., 2019d). These methods involve pipelines often trained and evaluated end-to-end, and surprisingly little work has been done to evaluate the layout generation component itself. Layout generation serves as a complementary task to these works and can be combined with these methods. In this work, we evaluate the layout modeling

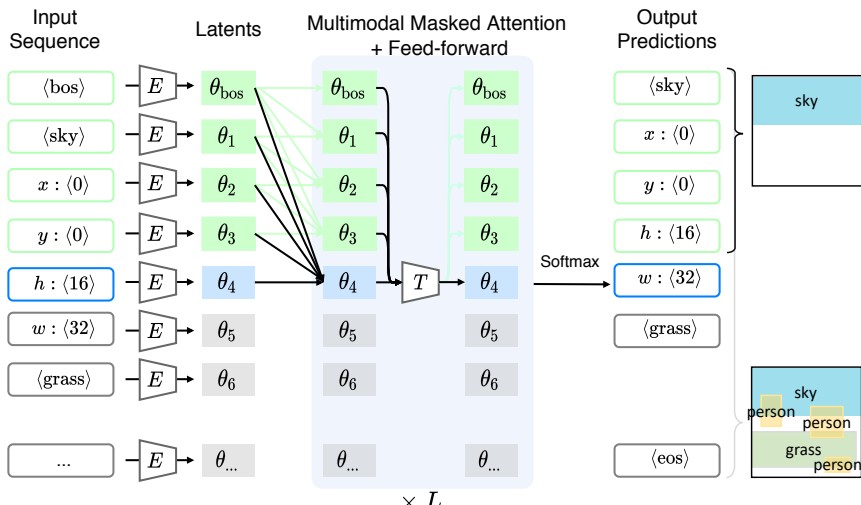

Figure 2: The architecture for MMA depicted for a toy example. It takes layout elements as input and predicts the next layout elements as output. During training, we use teacher forcing, *i.e.*, use the ground-truth layout tokens as input to a multi-head decoder block. The first layer of this block is a masked self-attention layer, which allows the model to see only the previous elements in order to predict the current element. We pad each layout with a special ⟨bos⟩ token in the beginning and ⟨eos⟩ token in the end. To generate new layouts, we perform nucleus sampling starting with just the ⟨bos⟩ token or a partial sequence.

capabilities of two of the recent works (Johnson et al., 2018; Li et al., 2019d) that have layout generation as an intermediate step. We also demonstrate the results of our model with Layout2Im (Zhao et al., 2019) for image generation.

**Layout generation.** The automatic generation of layouts is an important problem in graphic design. Many of the recent data-driven approaches use data specific constraints in order to model the layouts. For example, Wang et al. (2018a; 2019); Li et al. (2019c); Ritchie et al. (2019) generates top-down view indoor rooms layouts but make several assumptions regarding the presence of walls, roof *etc.*, and cannot be easily extended to other datasets. In this paper, we focus on approaches that have fewer domain-specific constraints. LayoutGAN (Li et al., 2019b) uses a GAN framework to generate semantic and geometric properties of a fixed number of scene elements. LayoutVAE (Jyothi et al., 2019) starts with a label set, *i.e.*, categories of all the elements present in the layout, and then generates a feasible layout of the scene. Zheng et al. (2019) attempt to generate document layouts given the images, keywords, and category of the document. Patil et al. (2019) proposes a method to construct hierarchies of document layouts using a recursive variational autoencoder and sample new hierarchies to generate new document layouts. Manandhar et al. (2020) develops an auto-encoding framework for layouts using Graph Networks. 3D-PRNN (Zou et al., 2017), PQ-Net (Wu et al., 2020) and ComplementMe Sung et al. (2017), generates 3D shapes via sequential part assembly. While 3D-PRNN generates only bounding boxes, PQ-Net and ComplementMe can synthesize complete 3D shapes starting with a partial or no input shape.

Our approach offers several advantages over current layout generation approaches without sacrificing their benefits. By factorizing primitives into structural parameters and compositional geometric parameters, we can generate high-resolution primitives using distributed representations and consequently, complete scenes. The autoregressive nature of the model allows us to generate layouts of arbitrary lengths as well as start with partial layouts. Further, modeling the position and size of primitives as discrete values (as discussed in §3.1) helps us realize better performance on datasets, such as documents and app wireframes, where bounding boxes of layouts are typically axis-aligned. We evaluate our method both quantitatively and qualitatively with state-of-the-art methods specific to each dataset and show competitive results in very diverse domains.

## 3   OUR APPROACH

In this section, we introduce our attention network in the context of the layout generation problem. We first discuss our representation of layouts for primitives belonging to different domains. Next,

we discuss the Multimodal Attention (MMA) framework and show how we can leverage previous advances such as Transformers (Vaswani et al., 2017) to model the probability distribution of layouts. MMA allows us to learn non-local semantic relationships between layout primitives and also gives us the flexibility to work with variable length layouts.

## 3.1 LAYOUT REPRESENTATION

Given a dataset of layouts, a single layout instance can be defined as a graph $\mathcal{G}$ with $n$ nodes, where each node $i \in \{1, \ldots, n\}$ is a graphical primitive. We assume that the graph is fully-connected, and let the attention network learn the relationship between nodes. The nodes can have structural or semantic information associated with them. For each node, we project the information associated with it to a $d_{\text{model}}$-dimensional space represented by feature vector $\mathbf{s}_i$. Note that the information itself can be discrete (*e.g.*, part category), continuous (*e.g.*, color), or multidimensional vectors (*e.g.*, signed distance function of the part) on some manifold. Specifically, in our ShapeNet experiments, we use an MLP to project part embedding to $d_{\text{model}}$-dimensional space, while in the 2D layout experiments, we use a learned $d_{\text{model}}$-dimensional category embedding which is equivalent to using an MLP with zero bias to project one-hot encoded category vectors to the latent space.

Each primitive also carries geometric information $\mathbf{g}_i$ which we factorize into a position vector and a scale vector. For the layouts in $\mathbb{R}^2$ such as images or documents, $\mathbf{g}_i = [x_i, y_i, h_i, w_i]$, where $(x, y)$ are the coordinates of the centroid of primitive and $(h, w)$ are the height and width of the bounding box containing the primitive, normalized with respect to the dimensions of the entire layout.

**Representing geometry with discrete variables.** We apply an 8-bit uniform quantization on each of the geometric fields and model them using Categorical distribution. Discretizing continuous signals is a practice adopted in previous works for image generation such as PixelCNN++ (Salimans et al., 2017), however, to the best of our knowledge, it has been unexplored in the layout modeling task. We observe that even though discretizing coordinates introduces approximation errors, it allows us to express arbitrary distributions which we find particularly important for layouts with strong symmetries such as documents and app wireframes. We project each geometric field of the primitive independently to the same $d_{\text{model}}$-dimension, such that $i^{\text{th}}$ primitive in $\mathbb{R}^2$ can be represented as $(\mathbf{s}_i, \mathbf{x}_i, \mathbf{y}_i, \mathbf{h}_i, \mathbf{w}_i)$. We concatenate all the elements in a flattened sequence of their parameters. We also append embeddings of two additional parameters $\mathbf{s}_{\langle\text{bos}\rangle}$ and $\mathbf{s}_{\langle\text{eos}\rangle}$ to denote start and end of sequence. Our layout in $\mathbb{R}^2$ can now be represented by a sequence of $5n + 2$ latent vectors

$$\mathcal{G} = (\mathbf{s}_{\langle\text{bos}\rangle}; \mathbf{s}_1; \mathbf{x}_1; \mathbf{y}_1; \mathbf{h}_1; \mathbf{w}_1; \ldots; \mathbf{s}_n; \mathbf{x}_n; \mathbf{y}_n; \mathbf{h}_n; \mathbf{w}_n; \mathbf{s}_{\langle\text{eos}\rangle})$$

For brevity, we use $\boldsymbol{\theta}_j$, $j \in \{1, \ldots, 5n + 2\}$ to represent any element in the above sequence. We can now pose the problem of modeling this joint distribution as product over series of conditional distributions using chain rule:

$$p(\boldsymbol{\theta}_{1:5n+2}) = \prod_{j=1}^{5n+2} p(\boldsymbol{\theta}_j | \boldsymbol{\theta}_{1:j-1}) \tag{1}$$

## 3.2 MODEL ARCHITECTURE AND TRAINING

Our overall architecture is surprisingly simple and shown in Fig. 2. Given an initial set of $K$ visible primitives (where $K$ can be 0 when generating from scratch), our attention based model takes as input, a random permutation of the visible nodes, $\boldsymbol{\pi} = (\pi_1, \ldots, \pi_K)$, and consequently a sequence of $d_{\text{model}}$-dimensional vectors $(\boldsymbol{\theta}_1, \ldots, \boldsymbol{\theta}_{5K})$. We find this to be an important step since by factorizing primitive representation into geometry and structure fields, our attention module can explicitly assign weights to individual coordinate dimensions. The attention module is similar to Transformer Decoder (Vaswani et al., 2017) and consists of $L$ attention layers, each of which consists of (a) a masked multi-head attention layer ($\mathbf{h}^{\text{attn}}$), and (b) fully connected feed forward layer ($\mathbf{h}^{\text{fc}}$). Each sublayer also adds residual connections (He et al., 2016) and LayerNorm (Ba et al., 2016).

$$\hat{\boldsymbol{\theta}}_j = \text{LayerNorm}(\boldsymbol{\theta}_j^{l-1} + \mathbf{h}^{\text{attn}}(\boldsymbol{\theta}_1^{l-1}, \ldots, \boldsymbol{\theta}_{5n+2}^{l-1})) \tag{2}$$

$$\boldsymbol{\theta}_j^l = \text{LayerNorm}(\hat{\boldsymbol{\theta}}_j + \mathbf{h}^{\text{fc}}(\hat{\boldsymbol{\theta}}_j)) \tag{3}$$

where $l$ denotes the layer index. Masking is performed such that $\boldsymbol{\theta}$ only attends to all the input latent vectors as well as previous predicted latent vectors. The output at the last layer corresponds to next

parameter. At training and validation time, we use teacher forcing, *i.e.*, instead of using output of previous step, we use groundtruth sequences to train our model efficiently.

**Loss.** We use a softmax layer to get probabilities if the next parameter is discrete. Instead of using a standard cross-entropy loss, we minimize KL-Divergence between softmax predictions and output one-hot distribution with Label Smoothing (Szegedy et al., 2016), which prevents the model from becoming overconfident. If the next parameter is continuous, we use an $L^1$ loss.

$$\mathcal{L} = \mathbb{E}_{\boldsymbol{\theta} \sim \text{Disc.}} [ \, D_{\text{KL}}(\text{SoftMax}(\boldsymbol{\theta}^L) \, \| \, p(\boldsymbol{\theta}')) \, ] + \lambda \mathbb{E}_{\boldsymbol{\theta} \sim \text{Cont.}} [ \, ||\boldsymbol{\theta} - \boldsymbol{\theta}'||_1 \, ] \tag{4}$$

**3D Primitive Auto-encoding.** PartNet dataset (Yu et al., 2019) consists of 3D objects decomposed into simpler meaningful primitives, such as chairs are composed of back, arms, 4 legs, and so on. We pose the problem of 3D shape generation as generating a layout of such primitives. We use Chen & Zhang (2019)'s approach to first encode voxel-based represent of primitive to $d_{\text{model}}$-dimensional latent space using 3D CNN. An MLP based implicit parameter decoder projects the latent vector to the surface occupancy grid of the primitive.

**Order of primitives.** One of the limitations of an autoregressive modeling approach is that sequence of primitives is an important consideration, in order to train the generative model, even if the layout doesn't have a natural defined order Vinyals et al. (2015). To generate a layout from any partial layout, we use a random permutation of primitives as input to the model. For the output, we always generate the sequences in raster order of centroid of primitives, *i.e.*, we order the primitives in ascending order of their $(x, y, z)$ coordinates. In our experiments, we observed that the ordering of elements is important for model training. Note that similar limitations are faced by contemporary works in layout generation (Jyothi et al., 2019; Li et al., 2019d; Hong et al., 2018; Wang et al., 2018a), image generation (Salimans et al., 2017; Gregor et al., 2015) and 3D shape generation (Wu et al., 2020; Zou et al., 2017). Generating a distribution over an order-invariant set of an arbitrary number of primitives is an exciting problem and we will explore it in future research.

**Other details.** In our base model, we use $d_{\text{model}} = 512$, $L = 6$, and $n_{\text{head}} = 8$ (number of multi-attention heads). Label smoothing uses an $\epsilon = 0.1$, and $\lambda = 1$. We use Adam optimizer (Kingma & Ba, 2014) with $\beta_1 = 0.9, \beta_2 = 0.99$ and learning rate $10^{-4}$ ($10^{-5}$ for PartNet). We use early stopping based on validation loss. In the ablation studies provided in § B, we show that our model is quite robust to these choices, as well as other hyperparameters (layout resolution, ordering of elements, ordering of fields). To sample a new layout, we can start off with just a start of sequence embedding or an initial set of primitives. Several decoding strategies are possible to recursively generate primitives from the initial set. In samples generated for this work, unless otherwise specified, we have used nucleus sampling (Holtzman et al., 2019), with top-$p = 0.9$ which has been shown to perform better as compared to greedy sampling and beam search (Steinbiss et al., 1994).

## 4 EXPERIMENTS

In this section, we discuss the qualitative and quantitative performance of our model on different datasets. Evaluation of generative models is hard, and most quantitative measures fail in providing a good measure of novelty and realism of data sampled from a generative model. We will use dataset-specific quantitative metrics used by various baseline approaches and discuss their limitations wherever applicable. We will provide the code and pretrained models to reproduce the experiments.

### 4.1 3D SHAPE SYNTHESIS (ON PARTNET DATASET)

PartNet is a large-scale dataset of common 3D shapes that are segmented into semantically meaningful parts. We use two of the largest categories of PartNet - Chairs and Lamp. We voxelize the shapes into $64^3$ and train an autoencoder to learn part embeddings similar to the procedure followed by PQ-Net (Wu et al., 2020). Overall, we had 6305 chairs and 1188 lamps in our datasets. We use the official train, validation, and test split from PartNet in our experiments. Although it is fairly trivial to extend our method to train for the class-conditional generation of shapes, in order to compare with baselines fairly, we train separate models for each of the categories.

**Generated Samples.** Fig. 3 shows examples of shape completion from the PartNet dataset. Given a random primitive, we use our model to iteratively predict the latent shape encoding of the next part,

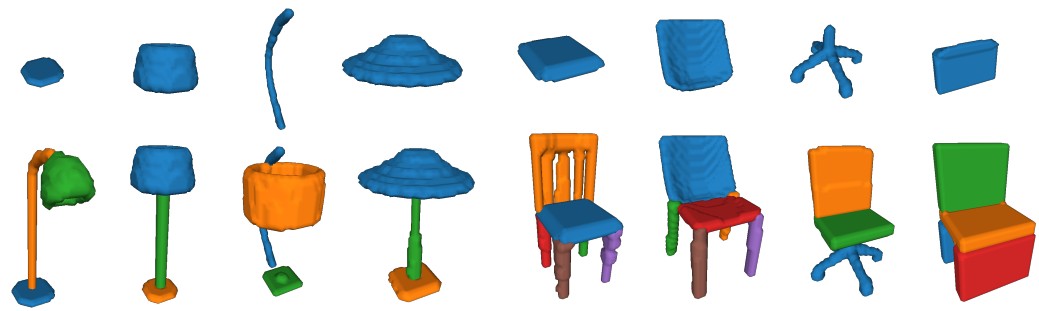

Figure 3: **Generated 3D objects.** Top row shows input primitives to the model. Bottom row shows the layout obtained with greedy decoding.

Table 1: Evaluation of generated shapes in Chair category.

| Method | JSD↓ | MMD↓ (CD) | MMD↓ (EMD) | Cov↑ (CD) | Cov↑ (MMD) | 1-NNA↓ (CD) | 1-NNA↓ (MMD) |
|---|---|---|---|---|---|---|---|
| PointFlow (Yang et al., 2019) | 1.74 | 2.42 | 7.87 | 46.83 | 46.98 | 60.88 | **59.89** |
| StructureNet (Mo et al., 2019) | 4.77 | 0.97 | 15.24 | 29.67 | 31.7 | 75.32 | 74.22 |
| IM-Net (Chen & Zhang, 2019) | 0.84 | **0.74** | 12.28 | 52.35 | 54.12 | 68.52 | 67.12 |
| PQ-Net (Wu et al., 2020) | **0.83** | 0.83 | 14.16 | 54.91 | **60.72** | 71.31 | 67.8 |
| Ours | 1.50 | 1.92 | **7.38** | **55.25** | 55.44 | **60.67** | 60.46 |

as well its position and scale in 3D. We then use the part decoder to sample points on the surface of the object. For visualization, we use the marching cubes algorithm to generate a mesh and render the mesh using a fixed camera viewpoint.

**Quantitative Evaluation.** The output of our model is point clouds sampled on the surface of the 3D shapes. We use Chamfer Distance (CD) and Earth Mover's Distance (EMD) to compare two point clouds. Following prior work, we use 4 different metrics to compare the distribution of shapes generated from the model and shapes in the test dataset: (i) Jensen Shannon Divergence (JSD) computes the KL divergence between marginal distribution of point clouds in generated set and test set, (ii) Coverage (Cov) - compares the distance between each point in generated set to its nearest neighbor in test set, (iii) Minimum Matching Distance (MMD) - computes the average distance of each point in test set to its nearest neighbor in generated set, and (iv) 1-nearest neighbor accuracy (1-NNA) uses a 1-NN classifier see if the nearest neighbor of a generated sample is coming from generated set or test set. Our model performs competitively with existing approaches to generate point clouds.

## 4.2 LAYOUTS FOR NATURAL SCENES (COCO BOUNDING BOXES)

COCO bounding boxes dataset is obtained using bounding box annotations in COCO Panoptic 2017 dataset (Lin et al., 2014). We ignore the images where the *isCrowd* flag is true following the LayoutVAE (Jyothi et al., 2019) approach. The bounding boxes come from all 80 thing and 91 stuff categories. Our final dataset has 118280 layouts from COCO train split with a median length of 42 elements and 5000 layouts from COCO valid split with a median length of 33. We use the official validation split from COCO as test set in our experiments, and use 5% of the training data as validation.

**Baseline Approaches.** We compare our work with 4 previous methods. **LayoutGAN** (Li et al., 2019b) is a GAN based layout generation framework, starting with a noise vector sampled from gaussian distribution to generate a bounding box layours. Since the method always generate fixed number of bounding boxes, it uses non-maximum suppression (NMS) to remove duplicates.

**LayoutVAE** (Jyothi et al., 2019) uses consists of two separate autoregressive VAE models. The method assumes categories of elements present in a generated layout to be known. First, CountVAE generates counts of each of the elements of the label set, and then BoundingBoxVAE, generates the location and size of each occurrence of the bounding box. **ObjGAN** (Li et al., 2019d) is a GAN framework for text to image synthesis. An intermediate step in their image synthesis approach is to generate a bounding box layout given a sentence using a BiLSTM (trained independently). We

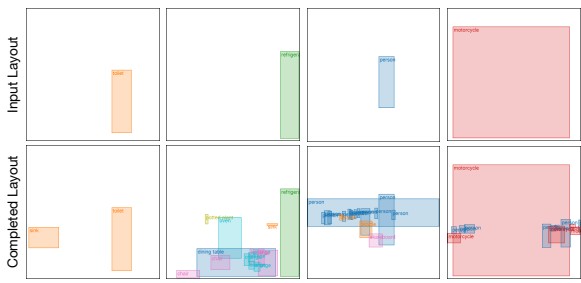

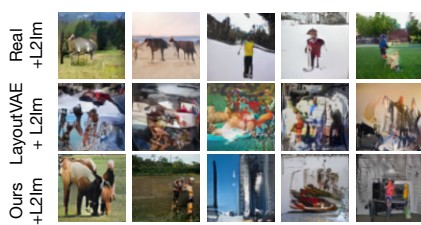

Figure 5: **Downstream task.** Image generation with layouts (Zhao et al., 2019).

Figure 4: **Generated layouts.** Top row shows seed layouts input to the model. Bottom row shows the layout obtained with greedy decoding. We skip the 'stuff' bounding boxes for clarity.

adopt this step of the ObjGAN framework to our problem setup by provide categories of all layout elements as input to the ObjGAN and synthesize all the elements' bounding boxes. **sg2im** (Johnson et al., 2018) attempts to generate images given scene graph of the image by first generating a layout of the scene using graph convolutions and then using the layout to generate complete scene using GANs. Since sg2im requires a scene graph input, following the approach of (Jyothi et al., 2019), we create a scene graph from the input and reproduce the input layout using the scene graph.

Since LayoutVAE and LayoutGAN are not open source, we implemented our own version of these baseline. Note that, like many GAN models, LayoutGAN was notoriously hard to train and our implementation (and hence results) might differ from author's implementation despite our best efforts. We were able to reproduce LayoutVAE's results on COCO dataset as proposed in the original paper and train our own models for different datasets using the same hyperparameters. We also re-purpose ObjGAN and sg2im using guidelines mentioned in LayoutVAE. Although evaluating generative models is challenging, we attempt to do a fair comparison to the best of our abilities. For our model (and others), we keep architecture hyperparameters same across the datasets. We also train different baselines for same number of epochs in corresponding datasets. Some of the ablation studies are provided in the appendix.

**Generated Samples.** Fig. 4 shows layout completion task using our model on COCO dataset. Although the model is trained with all 171 categories, in the figure we only show 'thing' categories for clarity. We also use the generated layouts for a downstream application of scene generation (Zhao et al., 2019).

**Semantics Emerge via Layout.** We posited earlier that capturing layout should capture contextual relationships between various elements. We provide further evidence of our argument in Fig. 6. We visualize the 2D-tsne plot of the learned embeddings for categories. We observe that super-categories from COCO are clustered together in the embedding space of the model. Certain categories such as window-blind and curtain (which belong to different super-categories) also appear close to each other. These observations are in line with observations made by Gupta et al. (2019) who use visual co-occurence to learn category embeddings.

**Quantitative evaluation.** Following the approach of LayoutVAE, we compute negative log-likelihoods (NLL) of all the layouts in validation data using importance sampling. NLL approach is good for evaluating validation samples, but fails for generated samples. Ideally, we would like to evaluate the performance of a generative model on a downstream task. To this end, we employ Layout2Im (Zhao et al., 2019) to generate an image from the layouts generated by each of the method. We compute Inception Score (IS) and Fréchet Inception Distance (FID) to compare quality and diversity of generated images. Our method is competitive with existing approaches in both these metrics, and outperforms existing approaches in terms of NLL.

Note that ObjGAN and LayoutVAE are conditioned on the label set. So we provide labels of objects present in the each validation layout as input. The task for the model is to then predict the number and postition of these objects. Hence, these methods have unfair advantage over our method and ObjGAN indeed performs better than our method and LayoutGAN, which are unconditional. We clearly outperform LayoutGAN on IS and FID metrics.

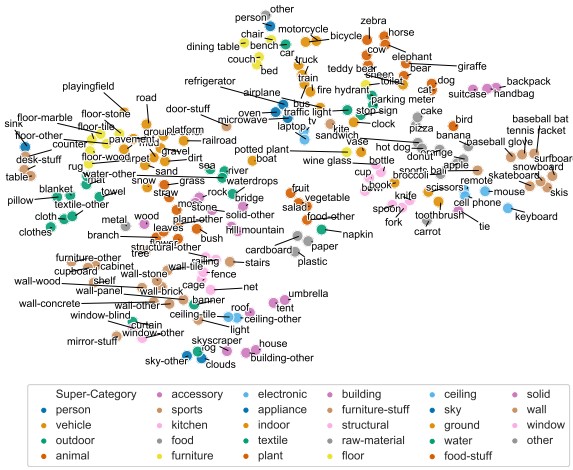

Figure 7: **Quantitative Evaluations on COCO.** Negative log-likelihood (NLL) of all the layouts in the validation set (lower the better). We use the importance sampling approach described in Jyothi et al. (2019) to compute. We also generated images from layout using Zhao et al. (2019) and compute IS and FID. Following Johnson et al. (2018), we randomly split test set samples into 5 groups and report standard deviation across the splits. The mean is reported using the combined test set.

| Model | NLL↓ | IS↑ | FID↓ |
|---|---|---|---|
| LayoutGAN (Li et al., 2019b) | - | 3.2 (0.22) | 89.6 (1.6) |
| LayoutVAE (Jyothi et al., 2019) | 3.29 | 7.1 (0.41) | 64.1 (3.8) |
| ObjGAN (Li et al., 2019d) | 5.24 | **7.5 (0.44)** | 62.3 (4.6) |
| sg2im (Johnson et al., 2018) | 3.4 | 3.3 (0.15) | 85.8 (1.6) |
| Ours | **2.28** | 7.1 (0.30) | **57.0 (3.5)** |

Figure 6: TSNE plot of learned category embeddings. Words are colored by their super-categories provided in the COCO. Observe that semantically similar categories cluster together. Cats and dogs are closer as compared to sheep, zebra, or cow.

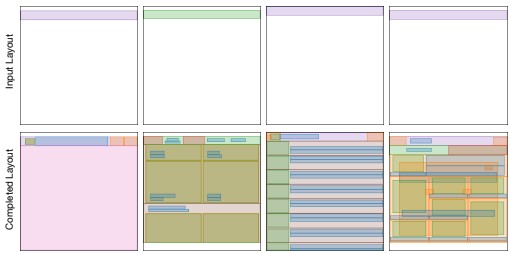

Figure 8: **RICO layouts.** Layouts obtained with greedy decoding for the RICO dataset. We skip the categories of bounding boxes for the sake of clarity.

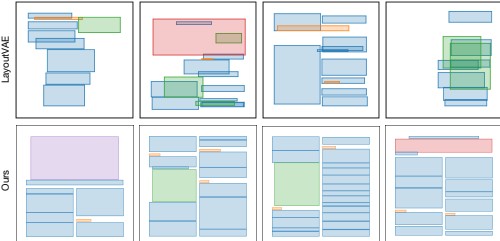

Figure 9: **Document Layouts.** Generated samples LayoutVAE (top) and our method (bottom). Our method produces aligned bounding boxes for various elements.

## 4.3 MOBILE APP WIREFRAMES (RICO) AND DOCUMENT LAYOUTS (PUBLAYNET)

**Rico Mobile App Wireframes.** Rico mobile app dataset (Deka et al., 2017; Liu et al., 2018) consists of layout information of more than 66000 unique UI screens from over 9300 android apps. Each layout consists of one or more of the 25 categories of graphical elements such as text, image, icon *etc.* A complete list of these elements is provided in the supplementary material. Overall, we get 62951 layouts in Rico with a median length of 36. Since the dataset doesn't have official splits, we use 5% of randomly selected layouts for validation and 15% for testing.

**PubLayNet.** PubLayNet (Zhong et al., 2019) is a large scale document dataset consisting of over 1.1 million articles collected from PubMed Central. The layouts are annotated with 5 element categories - text, title, list, label, and figure. We filter out the document layouts with over 128 elements. Our final dataset has 335703 layouts from PubLayNet train split with a median length of 33 elements and 11245 layouts from PubLayNet dev split with a median length of 36. We use the provided dev split as our test set and 5% of the training data for validation.

**Generated layout samples.** Fig. 8 and 9 shows some of the generated samples of our model from RICO mobile app wireframes and PubLayNet documents. Note that both the datasets share similarity in terms of distribution of elements, such as high coverage in terms of space, very little collision of elements, and most importantly alignment of the elements along both x and y-axes. Our method is able to preserve most of these properties as we discuss in the next section. Fig. 10 shows multiple completions done by our model for the same initial element.

**Comparison with baselines.** We use the same baselines for evaluation as discussed previously in §4.2. Fig. 9 shows that our method is able to preserve alignment between bounding boxes better

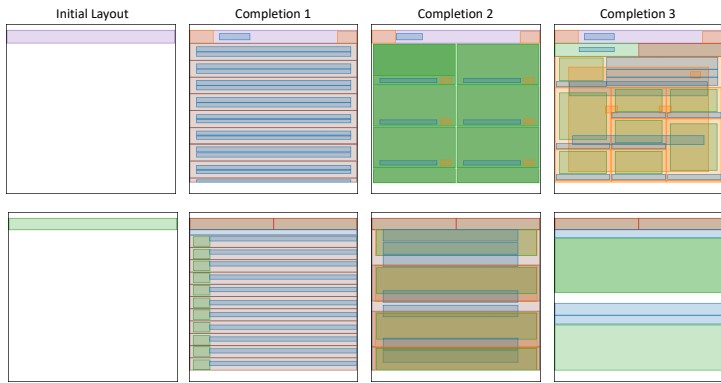

Figure 10: Multiple completions from same initial element

Table 2: Spatial distribution analysis for the samples generated using model trained on RICO and PubLayNet dataset. Closer the Overlap and Coverage values to real data, better is the performance. All values in the table are percentages (std in parenthesis)

| | RICO | | | PubLayNet | | |
|---|---|---|---|---|---|---|
| Methods | NLL↓ | Coverage | Overlap | NLL↓ | Coverage | Overlap. |
| sg2im (Johnson et al., 2018) | 7.43 | 25.2 (46) | 16.5 (31) | 7.12 | 30.2 (26) | 3.4 (12) |
| ObjGAN (Li et al., 2019d) | 4.21 | 39.2 (33) | 36.4 (29) | 4.20 | 38.9 (12) | 8.2 (7) |
| LayoutVAE (Jyothi et al., 2019) | 2.54 | 41.5 (29) | 34.1 (27) | 2.45 | 40.1 (11) | 14.5 (11) |
| LayoutGAN (Li et al., 2019b) | - | 37.3 (31) | 31.4 (32) | - | 45.3 (19) | 8.3 (10) |
| Ours | 1.07 | 33.6 (27) | 23.7 (33) | 1.10 | 47.0 (12) | 0.13 (1.5) |
| Real Data | - | 36.6 (27) | 22.4 (32) | - | 57.1 (10) | 0.1 (0.6) |

than competing methods. Note that we haven't used any post-processing in order to generate these layouts. Our hypothesis is that (1) discretization of size/position, and (2) decoupling geometric fields in the attention module, are particularly useful in datasets with aligned boxes.

To measure this performance quantitatively, we introduce 2 important statistics. **Overlap** represents the intersection over union (IoU) of various layout elements. Generally in these datasets, elements do not overap with each other and Overlap is small. **Coverage** indicates the percentage of canvas covered by the layout elements. Table 2 shows that layouts generated by our method resemble real data statistics better than LayoutGAN and LayoutVAE.

## 5 CONCLUSION.

We propose MMA, a multimodal attention framework to generate layouts of graphical elements. Our model uses self-attention model to capture contextual relationship between different layout elements and generate novel layouts, or complete partial layouts. We show that our model performs competitively with the state-of-the-art approaches for very diverse datasets such as Rico Mobile App Wireframes, COCO bounding boxes, PubLayNet documents, and 3D shapes. There are a few limitations of our approach. First, our model requires a layout or a scene to be decomposed into compositional primitives. In many cases, such primitives might not be even defined. Second, like most data-driven approaches, generated layouts are dominated by high frequency objects or shapes in the dataset. We can control the diversity to some extent using improved sampling techniques, however, generating diverse layouts that not only learn from data, but also from human priors or pre-defined rules is an important direction of research which we will continue to explore.

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

## Appendix

## A    Architecture and training details

In all our $\mathbb{R}^2$ experiments, our base model consists of $d_{\text{model}} = 512$, $L = 6$, $n_{\text{head}} = 8$, precision $= 8$ and $d_{\text{ff}} = 2048$. We also use a dropout of $0.1$ at the end of each feedforward layer for regularization. We fix the the maximum number of elements in each of the datasets to $128$ which covers over 99.9% of the layouts in each of the COCO, Rico and PubLayNet datasets. We also used Adam optimizer Kingma & Ba (2014) with initial learning rate of $10^{-4}$. We train our model for 300 epochs for each dataset with early stopping based on maximum log likelihood on validation layouts. Our COCO Bounding Boxes model takes about 1 day to train on a single NVIDIA GTX1080 GPU. Batching matters a lot to improve the training speed. We want to have evenly divided batches, with absolutely minimal padding. We sort the layouts by the number of elements and search over this sorted list to use find tight batches for training.

In all our $\mathbb{R}^3$ experiments, we change $d_{\text{model}} = 128$, and learning rate to $10^{-5}$.

## B    Ablation studies

We evaluate the importance of different model components with negative log-likelihood on COCO layouts. The ablation studies show the following:

**Small, medium and large elements:** NLL of our model for COCO large, medium, and small boxes is 2.4, 2.5, and 1.8 respectively. We observe that even though discretizing box coordinates introduces approximation errors, it later allows our model to be agnostic to large vs small objects.

**Varying precision:** Increasing it allows us to generate finer layouts but at the expense of a model with more parameters. Also, as we increase the precision, NLL increases, suggesting that we might need to train the model with more data to get similar performance (Table 3).

**Size of embedding:** Increasing the size of the embedding $d_{\text{model}}$ improves the NLL, but at the cost of increased number of parameters (Table 4).

**Model depth:** Increasing the depth of the model $L$, does not significantly improve the results (Table 5). We fix the $L = 6$ in all our experiments.

**Ordering of the elements:** Adding position encoding, makes the self-attention layer dependent to the ordering of elements. In order to make it depend less on the ordering of input elements, we take randomly permute the sequence. This also enables our model to be able to complete any partial layout. Since output is predicted sequentially, our model is not invariant to the order of output sequence also. In our experiments, we observed that predicting the elements in a simple raster scan order of their position improves the model performance both visually and in terms of negative log-likelihood. This is intuitive as filling the elements in a pre-defined order is an easier problem. We leave the task of optimal ordering of layout elements to generate layouts for future research. (Table 6).

**Discretization strategy:** Instead of the factorizing location in x-coordinates and y-coordinates, we tried predicting them at once (refer to the Split-xy column of Table 6). This increases the vocabulary size of the model (since we use $H \times H$ possible locations instead of $H$ alone) and in turn the number of hyper-parameters with decline in model performance. An upside of this approach is that generating new layouts takes less time as we have to make half as many predictions for each element of the layout (Table 6).

**Loss:** We tried two different losses, label smoothing (Müller et al., 2019) and NLL. Although optimizing using NLL gives better validation performance in terms of NLL (as is expected), we do not find much difference in the qualitative performance when using either loss function. (Table 6)

Table 3: Effect of $n_{\text{anchors}}$ on NLL

| $n_{\text{anchors}}$ | # params | COCO | Rico | PubLayNet |
|---|---|---|---|---|
| $32 \times 32$ | 19.2 | 2.28 | 1.07 | 1.10 |
| $8 \times 8$ | 19.1 | 1.69 | 0.98 | 0.88 |
| $16 \times 16$ | 19.2 | 1.97 | 1.03 | 0.95 |
| $64 \times 64$ | 19.3 | 2.67 | 1.26 | 1.28 |
| $128 \times 128$ | 19.6 | 3.12 | 1.44 | 1.46 |

Table 4: Effect of $d$ on NLL

| d | # params | COCO | Rico | PubLayNet |
|---|---|---|---|---|
| 512 | 19.2 | 2.28 | 1.07 | 1.10 |
| 32 | 0.8 | 2.51 | 1.56 | 1.26 |
| 64 | 1.7 | 2.43 | 1.40 | 1.19 |
| 128 | 3.6 | 2.37 | 1.29 | 1.57 |
| 256 | 8.1 | 2.32 | 1.20 | 1.56 |

Table 5: Effect of $L$ on NLL

| $L$ | # params | COCO | Rico | PubLayNet |
|---|---|---|---|---|
| 6 | 19.2 | 2.28 | 1.07 | 1.10 |
| 2 | 6.6 | 2.31 | 1.18 | 1.13 |
| 4 | 12.9 | 2.30 | 1.12 | 1.07 |
| 8 | 25.5 | 2.28 | 1.11 | 1.07 |

Table 6: Effect of other hyperparameters on NLL

| Order | Split-XY | Loss | # params | COCO | Rico | PubLayNet |
|---|---|---|---|---|---|---|
| raster | Yes | NLL | 19.2 | 2.28 | 1.07 | 1.10 |
| random | | | 19.2 | 2.68 | 1.76 | 1.46 |
| | No | | 21.2 | 3.74 | 2.12 | 1.87 |
| | | LS | 19.2 | 1.96 | 0.88 | 0.88 |

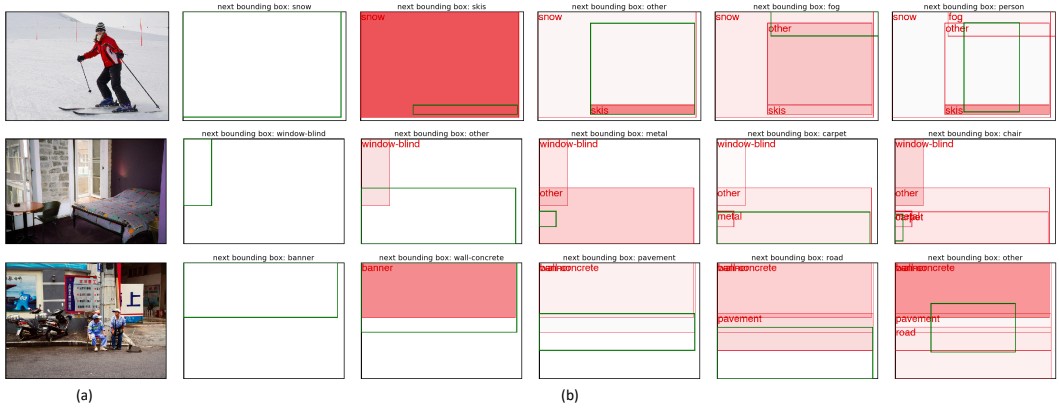

(a)      (b)

Figure 11: Visualizing attention. (a) Image source for the layout (b) In each row, the model is predicting one element at a time (shown in a green bounding box). While predicting that element, the model pays the most attention to previously predicted bounding boxes (in red). For example, in the first row, "snow" gets the highest attention score while predicting "skis". Similarly in the last column, "skis" get the highest attention while predicting "person".

## C    VISUALIZING ATTENTION

The self-attention based approach proposed enables us to visualize which existing elements are being attending to while the model is generating a new element. This is demonstrated in Figure 11

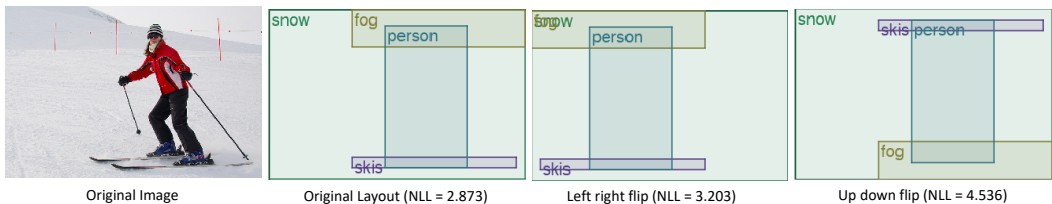

Original Image      Original Layout (NLL = 2.873)      Left right flip (NLL = 3.203)      Up down flip (NLL = 4.536)

Figure 12: We observe the impact of operations such as left right flip, and up down flip on log likelihood of the layout. We observe that unlikely layouts (such as fog at the bottom of image have higher NLL than the layouts from data.

Table 7: **Bigrams and trigrams**. We consider the most frequent pairs and triplets of (distinct) categories in real *vs.* generated layouts.

| Real | Ours | Real | Ours |
|------|------|------|------|
| other person | other person | person other person | other person clothes |
| person other | person clothes | other person clothes | person clothes tie |
| person clothes | clothes tie | person handbag person | tree grass other |
| clothes person | grass other | person clothes person | grass other person |
| chair person | other dining table | person chair person | wall-concrete other person |
| person chair | tree grass | chair person chair | grass other cow |
| sky-other tree | wall-concrete other | person other clothes | tree other person |
| car person | person other | person backpack person | person clothes person |
| person handbag | sky-other tree | person car person | other dining table table |
| handbag person | clothes person | person skis person | person other person |

Table 8: **Analogies**. We demonstrate linguistic nuances being captured by our category embeddings by attempting to solve word2vec (Mikolov et al., 2013) style analogies.

| Analogy | Nearest neighbors |
|---------|-------------------|
| snowboard:snow::surfboard:? | waterdrops, sea, sand |
| car:road::train:? | railroad, platform, gravel |
| sky-other:clouds::playingfield:? | net, cage, wall-panel |
| mouse:keyboard::spoon:? | knife, fork, oven |
| fruit:table::flower:? | potted plant, mirror-stuff |

## D  LAYOUT VERIFICATION

Since in our method it is straightforward to compute likelihood of a layout, we can use our method to test if a given layout is likely or not. Figure 12 shows the NLL given by our model by doing left-right and top-down inversion of layouts in COCO (following Li et al. (2019b)). In case of COCO, if we flip a layout left-right, we observe that layout remains likely, however flipping the layout upside decreases the likelihood (or increases the NLL of the layout). This is intuitive since it is unlikely to see fog in the bottom of an image, while skis on top of a person.

## E  MORE SEMANTICS IN LEARNED CATEGORY EMBEDDINGS

Table 7 captures the most frequent bigrams and trigrams (categories that co-occur) in real and synthesized layouts. Table 8 shows word2vec (Mikolov et al., 2013) style analogies being captured by embeddings learned by our model. Note that the model was trained to generate layouts and we did not specify any additional objective function for analogical reasoning task.

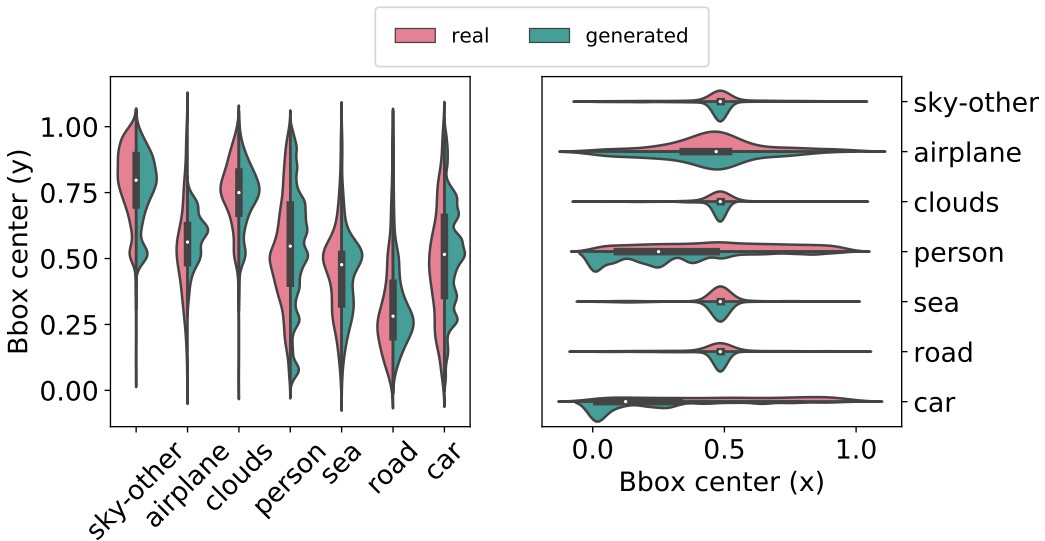

Figure 13: Distribution of xy-coordinates of bounding boxes centers. Distributions for generated layouts and real layouts is similar. The y-coordinate tends to be more informative (*e.g.*, sky on the top, road and sea at the bottom)

## F  DATASET STATISTICS

In this section, we share statistics of different elements and their categories in our dataset. In particular, we share the total number of occurrences of an element in the trai ning dataset (in descending

Table 9: Category statistics for Rico

| Category | # occurrences | # layouts | Category | # occurrences | # layouts |
|---|---|---|---|---|---|
| Text | 387457 | 50322 | Modal | 3248 | 3248 |
| Image | 179956 | 38958 | Pager Indicator | 2041 | 1528 |
| Icon | 160817 | 43380 | Slider | 1619 | 954 |
| Text Button | 118480 | 33908 | On/Off Switch | 1260 | 683 |
| List Item | 72255 | 9620 | Button Bar | 577 | 577 |
| Input | 18514 | 8532 | Toolbar | 444 | 395 |
| Card | 12873 | 3775 | Number Stepper | 369 | 147 |
| Web View | 10782 | 5808 | Multi-Tab | 284 | 275 |
| Radio Button | 4890 | 1258 | Date Picker | 230 | 217 |
| Drawer | 4138 | 4136 | Map View | 186 | 94 |
| Checkbox | 3734 | 1126 | Video | 168 | 144 |
| Advertisement | 3695 | 3365 | Bottom Navigation | 75 | 27 |

order) and the total number of distinct layouts an element was present in throughout the training data. Tables 9, 9 show the statistics for Rico wireframes, and table 10 show the statistics for PubLayNet documents.

Table 10: Category statistics for PubLayNet

| Category | # occurrences | # layouts |
|---|---|---|
| text | 2343356 | 334548 |
| title | 627125 | 255731 |
| figure | 109292 | 91968 |
| table | 102514 | 86460 |
| list | 80759 | 53049 |

## G  COORDINATE EMBEDDING

Just like in Fig. 6, we project the embedding learned by our model on COCO in a 2-d space using TSNE. In the absence of explicit constraints on the learned embedding, the model learns to cluster together all the coordinate embedding in a distinct space, in a ring-like manner.

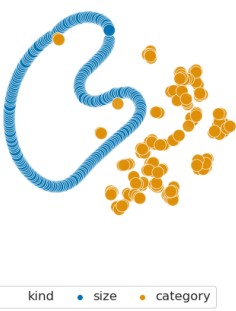

kind • size • category

Figure 14: TSNE plot for dimension embedding (256 of them) and category embedding for COCO.

## H    COMPARISON WITH POLYGEN

We would like to highlight some similarities of our framework with the recently proposed autoregressive generative model for 3D meshes, PolyGen Nash et al. (2020). While both works adopt Transformer Decoder for autoregressive modeling, the usage differs in the following aspects:

- PolyGen models mesh vertices as nodes. Advantage of this method is that it allows modelling high resolution 3D objects. However challenge is that sequence lengths for high resolution meshes can be very high and it can be very difficult to model them using self-attention (whose memory requirements grow proportionally to the square of sequence length).

- We on the other hand separate out attributes (not just coordinates but also height, width, category and (or) SDF encoding) of parts of 3D objects which are typically fewer in number. Deep Networks based SDF encoding are an active area of research and the current SOTA methods don't provide high resolution results as mesh based methods.

- Our model predicts future elements in order, but we randomize the order of the input elements. This allows us to do partial layout completion.

## I    NEAREST NEIGHBORS

To see if our model is memorizing the training dataset, we compute nearest neighbors of generated layouts using chamfer distance on top-left and bottom-right bounding box coordinates of layout elements. Figure 15 shows the nearest neighbors of some of the generated layouts from the training dataset. We note that nearest neighbor search for layouts is an active area of research.

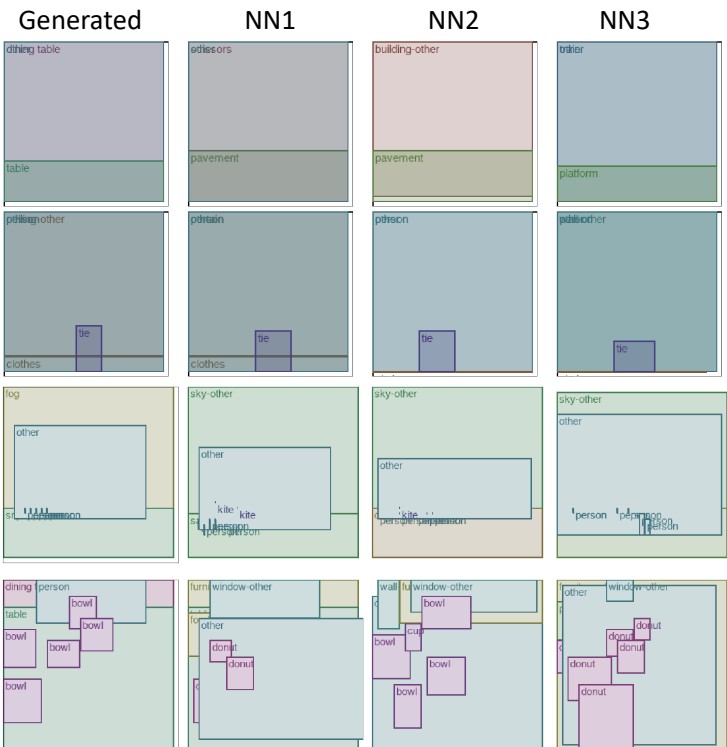

Figure 15: Nearest neighbors from training data. Column 1 shows samples generated by model. Column 2, 3, 4 show the 3 closest neighbors from training dataset. We use chamfer distance on bounding box coordinates to obtain the nearest neighbors from the dataset.

# J    MORE EXAMPLES FOR LAYOUT TO IMAGE

Layouts for natural scenes are cluttered and hard to qualitatively evaluate even for a trained user. Here we share some more sample layouts generated from two methods used in the paper. Figure 16 shows some extra sample layouts and corresponding images generated using Layout2Im tool. Existing layout to image methods don't work as well as free-form image generation methods but are arguably more beneficial in downstream applications. We hope that improving layout generation will aid the research community to develop better scene generation tools both in terms of diversity and quality.

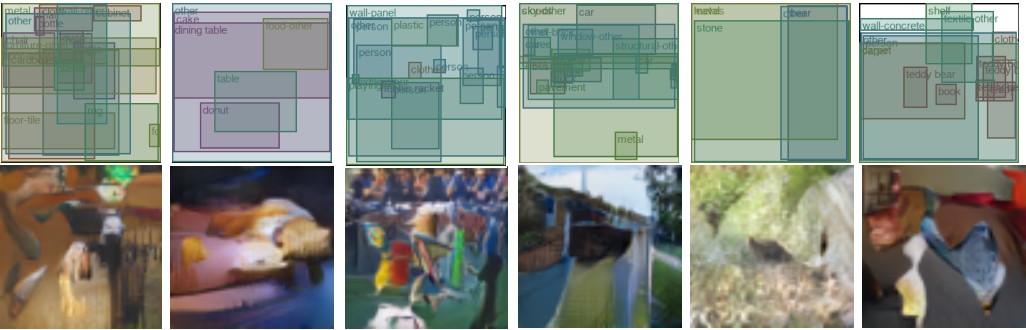

(a) LayoutVAE layouts (top) and images generated with Layout2Im (bottom)

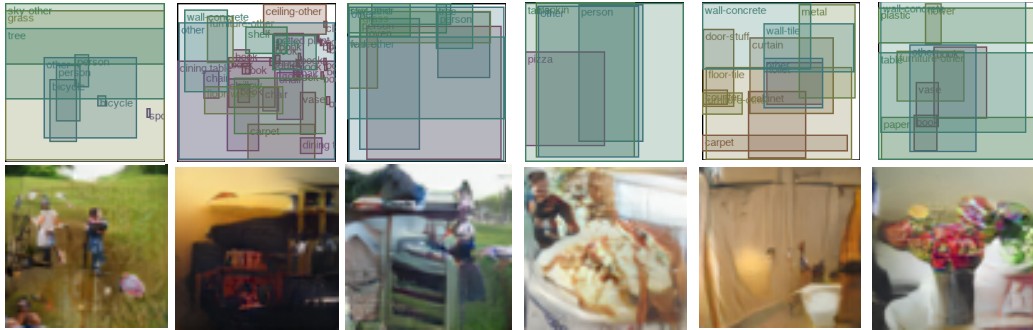

(b) Our layouts (top) and images generated with Layout2Im (bottom)

Figure 16: Some sample layouts and corresponding images

