# OpenReview forum: "Multimodal Attention for Layout Synthesis in Diverse Domains"
_ICLR.cc/2021/Conference — Reject_

### Official Review · AnonReviewer4 · 2020-10-26
**Results not state-of-the-art**

**Rating:** 5
**Confidence:** 4

**Review:**

The paper presents a transformer based architecture to model the probability distribution of scene layouts. The authors evaluate the model in four different application domains. The model performs competitively with state-of-the-art methods with respect to appropriate metrics.

**Strengths**
+ The paper is written clearly and the implementation details are appropriately described
+ The idea of using transformers is interesting
+ The evaluation of the method is thorough

**Weaknesses**
While the evaluation conducted by the authors is thorough, my main critique of this paper is that the results are not convincing enough to show the value of the proposed model.
- The quantitative results do not outperform the state-of-the-art models consistently across all metric. For instance, the mdoel does not outperform PointFlow and PQ-Net across all metrics in Table 1. Same is the case for ObjGAN (IS=7.5) vs proposed (IS=7.1) in Figure 7. The authors fail to comment on why is that the case.
- The qualitative samples are also not realistic in some cases. Some of the COCO results in Figure 4 and 5 do not look realistic -- layouts are too cluttered leading to incomprehensible scene. (row 2,col 3) in Figure 4 and (row 3,col 4) in Figure 5). From the layout samples presented in the paper, it seems the model produces cluttered layouts when the model is trying to generate higher number of objects.  The authors do not discuss this aspect in the paper. Is the quality of the layout dependent on the number of entities in the scenes? If yes, I think it is important for a scene layout generation model should be robust enough to handle flexible lengths.

Overall, while the paper has interesting approach to model the relations between the elements of a scene, the results are not convincing enough to demonstrate the effectiveness of the proposed model. Therefore, my initial rating is 5.

=========================================**Post-rebuttal comments**=======================================

I appreciate the revisions and additional visualization provided the authors. The revised version of the paper provides clarifications that make the paper easier to follow. While the paper presents an interesting idea, I am not convinced of the effectiveness of the proposed method based on the results provided by the authors. Therefore, my final rating as 5 (same as the initial rating).

---

> ### Author Response · Authors · 2020-11-12
> **Clarifications on results; paper revised**
>
> We thank you for your time and for acknowledging the clarity of the paper, interesting ideas, and thorough evaluation.
>
> One of the primary contributions of our work is to demonstrate how a powerful language model architecture with the right adaptations can be used to generate layouts in diverse domains. To the best of our knowledge, no other sequential part assembly framework performs competitively on diverse datasets.
>
> We intentionally keep the architecture similar to Vaswani et al's decoder, including several hyper-parameters, and propose to use node representations in a way suitable to represent layouts. Our approach allows the model to pay attention to the attributes that matter the most. This is in contrast with the commonly adopted practice of embedding all the attributes of the layout element together in a fixed latent space. While we tried multiple designs for self-attention, we observe that discretizing the position and size of elements combined with multimodal attention is good enough to achieve competitive performance across domains.
>
> We urge the reviewer not to discount a method that works robustly & achieves competitive results for missing a few benchmark numbers.
>
> 1) Our baseline methods ObjGAN and LayoutVAE are conditioned on the label set. So we provide input labels of objects present in each validation layout. The task for the model is to then predict the number and position of these objects. Hence, these methods have an unfair advantage over our method and LayoutGAN, which are unconditional. We clearly outperform LayoutGAN in these metrics
>
> 2) Indeed real-world scenes are cluttered and hard to evaluate qualitatively or with user studies on the layout task alone. This is the reason why we use an off-the-shelf layout-to-image generator for comparison. Layout-to-image methods don't work as well as free-form image generation methods yet. We have removed our claim that images from our method are better in Fig. 5 and leave it for the reader to decide which is better qualitatively. We provide quantitative comparisons in Fig 7. The median length of layouts in the COCO validation set is 33, our method is 25, and for LayoutVAE is 28 (although note that LayoutVAE starts with a label set).
> Does the quality of the image depend on the number of entities? Yes indeed, but it is the property of Layout2Im and an interesting direction for future research.
>
> We thank you again for these insightful observations. We have added the response to the first observation in the new revised version as well.

---

### Official Review · AnonReviewer2 · 2020-10-26
**Are the comparisons with the baselines fair?**

**Rating:** 5
**Confidence:** 4

**Review:**

### Summary

This work proposes a model to generate scene layouts by treating the scene as a composition of primitives, such as instance class, coordinates or scales. The model is a Transformer architecture, that attends on all previously predicted or given instance primitives. The probability of a scene layout is defined with a joint distribution, modeled as the product of conditional distributions using the chain rule. The model predicts an end of sequence token, that allows the generated layouts to have variable size. Moreover, the model allows to either complete an existing incomplete layout or to generate one from scratch. The paper presents experiments in four datasets, spanning different data domains, including 2D and 3D data.


### Strengths
+ The motivation and objective of this work are clear.
+ The general framework is simple and gives enough flexibility to apply it to different types of inputs and datasets.
+ Providing results on different data domain datasets helps supporting the work.

### Major concerns
Some important points of the method, experimental setup and the results are not clear (and need to be clarified), as they bring some important concerns:
- Using test set as validation set: It seems that the same split was used for both finding hyper-parameters (considering number of trained epochs due to early stopping also a hyper-parameter) and for reporting final results, at least for COCO-Stuff. It is not clear from this paper this is also the case for all the other datasets. In general, this is a bad practice and gives unfair advantage over baselines such as LayoutVAE, that uses a separate validation set to find hyper-parameters.
- Unclear comparison with baselines: Given that some metrics/datasets were not used in the original papers, I have some concerns regarding fairness of comparisons; (1) were the methods re-trained for this paper or taken as is from the code of original authors? (2) If the answer is the former, were hyper-parameters tuned independently for all datasets, as to give the chance for all baselines to do better on each given task? If the answer is the latter, it is unfair to use the same setup and hyper-parameters designed for one task and apply it for all.  (3) how were the proposed method's parameters tuned? are these parameters tuned for each dataset? Related to these questions, Figure 5 shows really bad results for LayoutVAE. It is unclear to me if this is what one would expect from this method and it makes me suspect the baseline is not properly used, as it seems as if this baseline is generating random layouts. It could help to see the generated layout for this baseline, and not only the final image.

### Other concerns
- In page 4, it is said that at validation time, teacher forcing is used. Does this mean that for the final reported results teacher forcing is used as well?  This is important to clarify, as always having access to the ground-truth layout is an unrealistic assumption at test time.
- The paper states in several parts that the framework allows to "generate a new layout either from an empty set or from an initial seed set of primitives". Which experiments show the method generating a layout from an empty set? It seems that the COCO-Stuff and 3D objects start from an initial set.  (1) Are the experiments on documents and applications from an empty set? (2) why not test the empty set start for COCO-Stuff and 3D objects? (3) when starting from an initial set, how many objects are used to start with? is it always the same number for all methods and setups?
- I would like to see standard deviations for all reported metrics, as it is unclear for some of them whether the position with respect to other baselines is meaningful or not.
- In Section 3 "other details", it is stated that nucleus sampling is used. However, throughout the paper, beam-search (Figure 2) or greedy-search (Figure 3) are mentioned. Which sampling method is actually used?


### Additional questions/comments
- As I understand, an EOS token is generated at some point and the layout generation stops. How is it prevented to generate a EOS token in the middle of an instance bounding box, let's say, after "h" and before outputting a "w" value. Is it only possible to generate an EOS token only when the next step generation is a class ID?

- At the end of Section 2, one of the listed advantages of the proposed method is "The autoregressive nature of the model allows us to generate layouts of arbitrary lengths as well as ...", while in fact other existing approaches (for instance, LayoutVAE) can also generate arbitrary length layouts, and it is not something introduced in this paper.

- In terms of general format:  Starting from the paragraph "3D Primitive Auto-encoding" in Page 5, this should go in another Section about experimental setup, not in the main method. Similarly, the conclusion should also be formatted as a separate section.

- Regarding the sentence: " The function that projects node i to latent space s_i can be learned independently or jointly with our layout generation framework." Where is this discussed in the paper or supported in experimental results? I could not find it and I am unsure of why is it mentioned here.
- Given that geometry coordinates (x,y, w,h) are discretized, which primitives remain continuous? Equation 4 contemplates the case where primitives are continuous, but it is not clear which primitives remain continuous after the discretization step.

- Why can't LayoutVAE and ObjGAN be applied to PoseNet? I would like to understand what is the limitations of these methods with respect to the proposed method.

### Reason for score
It is a simple approach based on existing work, just slightly adapted for this layout generation task. Moreover, I had to infer most of the experimental and comparison details, as it is not clear in the paper. Additionally, given the important concerns already mentioned above regarding the provided results, I cannot discern whether this simple approach brings any actual improvement over existing methods.

### What can be improved
- A clearer discussion of what are the advantages of the proposed method. Although other methods did not provide results on such diverse data domains, it seems that LayoutVAE and ObjGAN were adapted as baselines for all except the 3D dataset. Therefore, the flexibility to different data domains of the proposed method, a major selling point, could in fact also be present in other methods. If baselines are allowed to tune parameters for each specific dataset, do they provide consistently worse results than the proposed method?
- I would like the authors to extensively discuss the "Major concerns" and paint a better picture of the experimental setup. In this line, these details should also be included in the paper, whether it is in the main body or in the appendix.
- If all other concerns raised are solved, this could also be clarified in the text.
- Include results as mean and stdev over different random seeds for all experiments, providing more robust comparisons.


### After authors response
Although authors have addressed one of my main concerns and some minor ones, I still have doubts regarding the fairness of comparison with the baselines (lack of proper hyper-parameter optimization) and therefore cannot trust the results. All in all, I keep my rating.

---

> ### Author Response · Authors · 2020-11-12
> **Paper updated to include important clarifications regarding experimental setup**
>
> We thank you for your time, detailed feedback and acknowledging the simplicity and clarity of the paper, and flexibility of the framework that can be applied across multiple domains.
>
> Please allow us to highlight the advantages of the proposed method. Our work demonstrates how a powerful language architecture with few adaptations can be used to generate layouts in diverse domains. We intentionally keep the architecture similar to Vaswani et al's decoder and propose to use node representations in a way suitable to represent layouts. Multimodal attention allows the model to pay attention to the attributes that matter the most. This is in contrast with the common practice of embedding all the attributes of a layout element together in a fixed latent space. While we tried multiple designs for self-attention (separate K,Q,V matrices for attributes, learned position encodings), we observe that discretizing the position and size of elements combined with multimodal attention is good enough to achieve competitive performance across domains.
>
> We next address your major concerns (we agree these are important points and we have also revised the paper accordingly).
>
> Thank you for pointing out the important missing detail in the paper. We do not use the test set for validation (for early stopping or any other purpose). We strongly deplore such a practice and apologize for not clarifying this in the paper. We realize that we didn't specify that "val" set in official COCO dataset and "dev" set in official PubLayNet is used only for testing. We use official train/val/test split for Shapenet, 5% of the training set for validation in COCO, and PubLayNet (this is slightly different from LayoutVAE who uses 5000 training images for validation). RICO doesn't have an official split and we use 5% of data for validation and 15% for testing.
>
> **Reg. baselines**
> 1) We had to re-implement LayoutVAE and LayoutGAN since the codes weren't open-sourced. For ObjGAN and sg2Im, we used the official code and retrained the models. For ShapeNet experiments, we use the released models and don't train our own models.
> 2) and 3) We do not perform hyperparameter tuning for all datasets for all baselines, but we try to do a fair comparison to the best of our ability. To this end (i) we use the same hyperparameters for our model across datasets and (ii) we train different models for an equal number of epochs on different datasets. In fact, we keep most of the hyperparameters the same as in the original Transformer paper. Some of the ablation studies are provided in the appendix.
> Regarding figure 5 - layout to image methods don't work as well as free-form image generation methods yet. We have removed our claim that images from our method are better and leave it for the reader to decide which is better qualitatively. Quantitative comparisons are in Fig 7.
>
> **Other concerns**
> - In order to compute NLL, we use teacher forcing in the validation set (NLL is computed for each sample given model). We are not aware if there are other ways to compute NLL for validation data. All other statistics are provided by generating 1000 unconditional random samples from the dataset
> - Some experiments such as Fig. 9 show samples from an empty set. In all completion experiments in the paper, we have started with all attributes of one element. We have added multiple completions from the same initial set in Fig. 10.
> - Due to the limitation of computational resources, we won't be able to add the necessary standard deviations by end of the review period. We will try to add them before the final version is due.
> - We have used nucleus sampling in all experiments unless otherwise specified. The caption below Fig. 2 is now corrected.
>
> **Response to additional comments**
> 1) We stop the layout generation whenever an EOS token is generated and discard the elements for whom all attributes are not sampled
> 2) We agree that this is not something introduced by our work, and is a strength of our as well as other approaches (LayoutVAE, ObjGAN, PQ-Net)
> 3) Primitive representation further clarified in the first couple of paragraphs of Section 3.
> 4) In ShapeNet experiments, the part structure is represented by a continuous latent vector (learned independently during primitive autoencoding) and is not categorical. In all other datasets, a category is used which is discrete. Other attributes of primitives such as x,y,w also remain discrete
> 5) It is not straightforward to extend baseline methods to PartNet. For example, LayoutVAE starts with a label set of categories, ObjGAN starts with a text as input (which is not available in PartNet).
>
> **Updates to the paper**
>
> 1) Clarifications regarding train/val/test splits for different datasets
> 2) Additional details regarding our baseline implementations
> 3) Edited the caption of Fig. 2 and Fig. 5
> 4) Edited the description of Layout element representation in Section 3
> 5) Added multiple completions from the same initial set (with one element) in Fig. 10.

---

> > ### Comment · AnonReviewer2 · 2020-11-12
> > **Missing fair comparison to extract robust conclusions**
> >
> > I thank the authors for the clarifications provided.
> >
> > Regarding the comparison with baselines, unfortunately, I do not think the comparisons are fair without tuning each method properly with a hyper-parameter search for each dataset. Using the same hyper-parameters and number of epochs (how was this number chosen?) across datasets and methods is no guarantee for fair comparisons.
> > Additionally, in Appendix A you mention that you carefully construct the batches in a specific way. Do you use the same data batch techniques for all baselines or only for the proposed model?
> > My concern in Figure 5 is not due to the final generated image, but for the quality of generated layouts from LayoutVAE. I would like to see both the generated layouts and generated final image for all methods in Figure 5.
> >
> > I believe a more robust and fair experimental section would be a great addition to this paper, including the comments above and reporting mean and standard deviations across several runs.

---

> > > ### Author Response · Authors · 2020-11-13
> > > **Added more generated samples for COCO; clarifications**
> > >
> > > We have added Appendix J sharing more layout samples and images generated from those layouts. Also following [3]’s protocol, we randomly split test set samples into 5 groups and reported standard deviation across the splits in Figure 7. The mean is reported using the combined test set as before. We completely agree that our paper will be stronger once we add the standard deviations on multiple runs and we will add these numbers and release the code before the conference happens.
> > >
> > > We want to emphasize that the evaluation of generative models is a challenging task and an important direction for future research. Quantitative metrics included in our work (as well as previous works) are by no means an absolute test of the layout generation capabilities of any model. For instance,
> > > - NLL evaluates validation samples well but fails for generated samples (a model can generate low-quality samples with high probability)
> > > - FID, IS metrics convey a comparison of the distribution of generated image statistics with imagenet pre-trained inception model but they are hardly indicative of quality/diversity of individual samples.
> > > - Statistics such as coverage/overlap indicate the distribution of elements in samples but don't convey samples' quality
> > > - A carefully designed large scale user study can mitigate quality test but can be expensive and doesn't convey information regarding diversity
> > > - Point cloud generation metrics provide good numbers even if the model is just producing an average shape (as pointed out in PointFlow)
> > >
> > > That being said, please allow us to motivate you why ideas presented in our paper can be a valuable addition for the research community (regardless of whether or not our method is better than a baseline method in one or two evaluation metrics).
> > > A number of existing methods for layout modeling and synthesis attempt to represent and (or) generate layout element(s) in one shot. That is, a common practice [1, 2, 3, 4, 5] is to combine the representation of different attributes of a layout element in a single continuous vector and add an MLP/FC layer before or after this vector, based on whether one is encoding the information or decoding it, respectively. In contrast, we propose to separate out multiple attributes and let the model represent them as separate entities in a self-attention model. While generating, we can again sample attributes one at a time, while simultaneously paying attention to previous attributes that matter the most (as also observed by R1). Also, a decoder only design allows us to complete partial layouts. Experiments in our paper are aimed to demonstrate the point that this approach provides competitive results in 4 diverse real-world domains. And to the best of our knowledge, not trivial to demonstrate using existing methods. These experiments are by no means exhaustive but we believe that our approach can still provide a good starting point for a user starting out in a new domain, as well as in the domains considered in the paper.
> > >
> > >
> > > 1. Li, Jianan, et al. "Layoutgan: Generating graphic layouts with wireframe discriminators." ICLR 2019.
> > > 2. Jyothi, Akash Abdu, et al. "Layoutvae: Stochastic scene layout generation from a label set." ICCV 2019.
> > > 3. Johnson, Justin, Agrim Gupta, and Li Fei-Fei. "Image generation from scene graphs." CVPR 2018.
> > > 4. Li, Wenbo, et al. "Object-driven text-to-image synthesis via adversarial training."  CVPR 2019.
> > > 5. Wu, Rundi, et al. "PQ-NET: A generative part seq2seq network for 3D shapes." CVPR 2020.

---

### Official Review · AnonReviewer1 · 2020-10-28
**Good Message and Empirical Performance, but Rather Limited Novelty**

**Rating:** 6
**Confidence:** 4

**Review:**

=====Post-Rebuttal Comment=====

I thank the authors for their detailed response to my concerns.

While my opinion of this work remains largely similar, I raised by score from 5 to 6 for the following reasons:

- I do buy the argument that the proposed method "allow both developers and researchers to start from a strong method with a low barrier to entry in diverse domains".
- I do not share the concerns of R2 & R4 regarding the quality of results and fairness of comparison. As my primary concern (amount of technical innovation) is not shared by other reviewers, I am swayed to change my initial "on the borderline" rating to the positive side.

I still would recommend adding the following evaluations:

- more diverse initial state for Figure 10.
- a more interpretable dataset for Figure 15: I think ShapeNet would demonstrate this point better.


=====Summary=====

This work introduces a general framework for generating scene layouts in 2D or 3D. The key idea of this work is that one can represent an arbitrary layout as a sequence of objects, where each object contains a set of attributes (location, sizes, category, shape, etc.). By projecting these features into the same (high dimensional) embedding space, one can thus represent the entire layout as a sequence of embeddings. Consequently, it is possible to borrow architectures commonly used in natural language processing and design a generative model for such sequences. Here, the authors use a simple model with masked self attention and trained with teacher forcing. The proposed model is shown to have performance comparable or slightly better than SoTA methods in four different tasks under multiple metrics. It is also demonstrated that the learned embedded space displays some structure that respects the semantics.

=====Strengths=====
- The idea behind this paper seems to be general enough and can be applied to any scenarios where one can represent each entity in the layout share the same set of attributes.
- I like the idea of separating different attributes a lot --- probably my favorite part of the paper. It makes sense intuitively that doing so will allow the attention module to more easily focus on the attributes that matter.
- Strong empirical performance: comparable results for 3D shape synthesis and seems to achieve slightly better performances for the other three tasks evaluated.
- Clear exposition and comprehensive discussion of related works in multiple areas

=====Weaknesses=====
- Would like to mention that one of the main novelty (to my understanding) in this work has already been attempted in an earlier work “PolyGen: An Autoregressive Generative Model of 3D Meshes”. The vertex model of PolyGen, in particular, uses a similar transformer decoder to generate the positions of points, a task that is very similar to predicting the position of objects in the layout. (the coordinates are also discretized in a similar way to this work, and the ordering strategy is similar, but these are much more minor points).
- Following previous point: I think the novelty in this work is quite limited in general, using existing architectures and relying on ideas that have been explored (albeit in slightly different settings) before.
- The empirical performance of the method is not strong enough to convince me that it can replace the more domain specific methods compared here. The reason is two-fold. First, it is unclear to me whether the proposed method is flexible enough to handle all the other tasks that those domain-specific method could, as only generation results are shown here. Second, I am not sure if the proposed method is flexible enough / has the right inductive bias for the layout tasks. For example, not taking hierarchy into account seems to be a minus for me when one want to interpret / manipulate the outputs. The sequential nature of the model also put constraints on the type of partial input acceptable e.g. it seems to be hard to handle input objects with only partial information available)

=====Reasons for Score=====

I’m right on the borderline for this paper and think this can go either way.

On the positive side, the paper shows a very clear message (again) that models borrowed from NLP can be surprisingly effective if we find a way to convert other structures (layout here) to a sequence. The empirical performance is also quite decent. On the negative side, the amount of novelty behind the main message of this paper is questionable: neither the architecture nor the idea of converting stuff to sequences are completely new, and the empirical performance is not strong enough for one to favor this approach over others (in their respective tasks).

To me, the question boils down to whether the main message of this paper is a very important one that deserves to be heard more by the broader community. I am not particularly convinced here, and lean slightly towards rejecting this paper.

=====Additional Comments & Questions=====

Other minor questions in addition to what I listed above:

- I am not sure if it makes sense to map different types of features onto the same embedding space e.g. I don’t see how spatial coordinates and color can share any feature, and they would ideally just occupy distinct regions of the embedding space. Could the authors explain the design choice here? Why can’t one use different embedding spaces (and modify the attention module to handle that)?
- Would like to see examples demonstrating that the model can generate diverse outputs, as opposed to always do the same thing for the same (partial) input. The shape completion tasks in figure 3 seems to be a good task to do this on. (The authors mentioned in conclusion that diversity is a problem, but I would still want to see how much of a problem it is, as I think some of the other works compared here don’t suffer too much from diversity problems)
- Would also like to see a figure showing the nearest neighbor from the training set(s) to make sure that the model is not simply memorizing the input data.
- It is mentioned in section 3.1 that discretization helps learning symmetries, are there concrete evidence towards this?
- Figure 6: I get that there is structure here, but does the method learn better structure than other works?
- Figure 5: I don’t get the message of this figure - all the images look equally bad to me.

---

> ### Author Response · Authors · 2020-11-12
> **Addressed comments; Added clarifications and additional results for multiple completions from same input; nearest neighbors**
>
> We thank you for the time and in-depth feedback. Below we address the concerns in detail.
>
>
> **Comparison with Polygen**
> We agree with the reviewer that comparison with the vertex decoder of Polygen is relevant since both are autoregressive models with the transformer backbone and model individual coordinate dimensions separately. However, there are some important differences such as - (i) PolyGen models mesh vertices as nodes. The advantage of this approach is that it allows for modeling high-resolution 3D objects. However, the challenge is that sequence lengths for high-resolution meshes can be very high and it can be very difficult to model them using self-attention (whose memory requirements grow proportionally to the square of sequence length) (ii) We on the other hand separate out attributes (not just coordinates but also height, width, category and (or) SDF encoding) of parts of 3D objects which are typically fewer in number. Deep Networks based SDF encoding is an active area of research and the jury is still out on the better way to represent 3D models (iii) Our model predicts future elements in order, but we randomize the order of the input elements. This allows us to do partial layout completion.
> We have added this discussion in Appendix H for the benefit of the reader.
>
> **Regarding domain-specific methods**
> Domain-specific methods are an important part of a number of AI-assisted design applications. Our focus, in this work, however, is to develop a general-purpose layout generation framework that can be adapted to a new domain with minimal inductive biases. Our model analysis such as TSNE embeddings of categories shows that some of the biases present in the data are automatically learned by the model. We agree that accepting arbitrary information of input elements is a very interesting suggestion and an important direction of future research.
>
> As you said, one of the primary contributions of our work is to demonstrate how a powerful language model architecture with few adaptations can be used to generate layouts in diverse domains. One of our main goals while developing our work was to allow both developers and researchers to start from a strong method with a low barrier to entry in diverse domains. To this end, we intentionally keep the architecture similar to Vaswani et al's decoder, including several hyper-parameters, and propose to use node representations in a way suitable to represent layouts.
>
> **Regarding other minor concerns**
> 1)  Mapping different attributes to the same embedding space - Thank you for highlighting this. $d_{model}$ = 512 is a pretty large size for latent space and indeed the model learns to cluster together all the coordinate embeddings in a distinct space, in a ring-like manner (refer to newly added Appendix G). We tried two variations on this (1) separate query, key, value matrices for different attributes (2) Different positional encodings. None of the variations provided observable benefit on the proposed approach and we exclude those experiments for brevity.
> 2) Added multiple completions from the same layout (Figure 10, Section 4.3)
> 3) Added nearest neighbors from the training set for generated samples (Appendix I)
> 4) It is non-trivial to have an ablation study for discretization alone (see our response to R3), we do show in Figure 9 the apparent difference between layouts generated by models using continuous coordinates vs our method
> 5) The goal of figure 6 is to analyze the embeddings (and inherent structure in data) learned by model and not to beat embeddings learned in domains such as language. Understanding and evaluating the structure encoded in the embeddings, except for some tasks like retrieval, is difficult and an interesting direction for future research.
> 6) Figure 5 - Indeed, layout to image methods don't work as well as free-form image generation methods yet. We remove our claim that images from our method is better and leave it for the reader to decide which is better qualitatively. We provide quantitative comparisons in Fig 7.
>
> **Summary of changes** in the updated paper based on reviewer feedback
> 1) Added Appendix H to include a comparison with PolyGen
> 2) Added a new figure for coordinate embedding in Appendix G
> 3) Added multiple completions from the same layout (Figure 10, Section 4.3)
> 4) Added the nearest neighbors from the training set for generated samples (Appendix I)
> 5) Modified caption of Figure 5

---

### Official Review · AnonReviewer3 · 2020-10-29
**Clean architecture and good cross-domain applicability**

**Rating:** 7
**Confidence:** 3

**Review:**

This paper presents an auto-regressive method for generating layouts by sequentially synthesizing new elements. The architecture is not dramatically new, but it is well-justified and analyzed, and there are some interesting tweaks. The results are strongest in that they show good performance of essentially the same architecture and hyperpameters across quite different domains: to my knowledge such variety has not really been demonstrated for any of the assembly-based generative models I'm familiar with.

The discretization aspect is not completely clear. As I understand it, each scalar geometric attribute of a node (but not the feature vector s_i) in the layout (x_i, y_i, w_i, h_i) is quantized to 8 bits, so that it can be thought of as selecting from one of 256 discrete options. How is the projection to d_model dimensions then performed, if d_model is 512 in one domain and 128 in another? What is the "categorical distribution" referred to here? Could you please explain this section more clearly?

Also, would it be possible to do an ablation study where no discretization is done at all?

The following paper is relevant as a baseline which also does sequential part assembly for shape generation. It would be nice if the authors could compare to this work.

Sung et al., "ComplementMe: Weakly-Supervised Component Suggestion for 3D Modeling", SIGGRAPH Asia 2017.

This is another paper that presents a fairly general framework for an autoregressive layout generator. The current paper should be compared with this as well, via experiments if possible:

Ritchie et al., "Fast and Flexible Indoor Scene Synthesis via Deep Convolutional Generative Models", CVPR 2019

And this paper is more recent than the ones compared to in the paper for document layout generation:

Gadi Patil et al., "READ: Recursive Autoencoders for Document Layout Generation", CVPR 2020 Workshop on Text and Documents in Deep Learning Era.

(Various flavours of layout generation have seen a huge amount of research in the last few years, so I am not totally confident of my assessment especially vis-a-vis prior work.)

There are minor typos and grammatical errors -- a thorough proofread would help.

---

> ### Author Response · Authors · 2020-11-12
> **Paper updated to include clarifications; comments regarding discretization**
>
> Thank you for your time and for appreciating our paper. Indeed it was one of the primary goals of the paper to develop a layout generation framework with cross-domain applicability and to the best of our knowledge, we haven't come across such assembly-based generative models in the literature.
>
> **Regarding discretization**
> You're correct, the scalar attribute can be represented by one hot encoded 256-dimensional vector, and we use a 256x512 dimensional embedding matrix to project it to d_model space which is can also be thought of as using a 256x512 FC layer. s_i which is 128 dimensional (for 3D shapes) is projected to d_model space also using an FC layer (128x512). We have added this clarification in Section 3 in the revised version of the paper. Also, refer to a newly added figure in Appendix G for visualization of coordinate embedding.
>
> Ablation with no discretization - In our earlier experiments, where we had an encoder-decoder framework (instead of decoder only framework), we observed that having continuous outputs makes the model training hard in terms of convergence. Also, if we predict continuous values for coordinates/dimensions of elements, it is non-trivial to sample from them which makes apple to apple comparison with the discrete case difficult. That is why in many generative modeling works (LayoutGAN, LayoutVAE, PQ-Net), sampling is done for the entire layout at the beginning of the decoder. Please let us know if there is a specific modification you'd like us to try and we will be happy to oblige.
>
> **Relevant related works** - We thank you for pointing out very relevant related works. We have included each of them in the discussion in Section 2 in the revised version of the paper.
>
> Sung et al. - The setup of Sung et al. is similar to our problem setup for 3D shapes. Since the authors have a different data preprocessing pipeline to create parts, the results reported in their paper (on shape completion) are not directly comparable. However, we would try to extend our approach to their released dataset and compare the results. We do include PQ-Net and Structure-Net which were also part assembly frameworks and published after Sung et al.
>
> Ritchie et al. - This work utilizes several domain-specific constraints and it is non-trivial to extend them to our use-cases. For example, it assumes a top-down axes aligned image of the room and predicts first-tier and second-tier objects in the room based on "support".
>
> Patil et al. - This work uses spatial relationships such as "right, left, bottom, bottom-left, bottom-right, enclosed, and wide-bottom" to represent document hierarchies and proposes to use RvNN-VAE to learn and sample from them. READ doesn't open source the code and the larger document dataset used in their paper. It would be infeasible for us to reproduce their results and include them in our work during the review period.
>
> Kindly note that we did not mean to imply that having domain-specific constraints is a negative thing. In fact, many real-world applications for AI-assisted design would benefit from relevant constraints. Our focus, in this work, however, is to develop a general-purpose layout generation framework that can be adapted to a new domain with minimal knowledge.
>
>
> **Summary of changes** in the updated paper based on reviewer feedback
> 1) Clarification regarding node representation in Section 3.1
> 2) Appendix G added for visualization of coordinate embedding
> 3) A more detailed comparison with some of the baselines in Section 2

---

### Author Response · Authors · 2020-11-20
**Summary of changes; revised manuscript**

Dear reviewers and AC,

We thank you for your time and valuable suggestions. We have uploaded a new version of the paper with changes we discussed in our individual responses below. In particular, we
* clarified some of the notations, image captions, dataset splits, and baseline implementations
* included further discussion of related works
* added more samples corresponding to multiple completions from the same input sequence (Figure 10)
* added more samples generated from scratch (Appendix J)
* added nearest neighbors for generated samples (Appendix I)
* reported standard deviation for IS and FID across multiple splits in Figure 7
* added visualization of coordinate embedding (Appendix G)

We believe the reviews and updates have definitely helped us improve our submission. Given the time and computational constraints, we were not able to perform all possible experiments we intended, but we’ll submit at least one more revision before the final paper is due

Thank you

---

### Decision · Program_Chairs · 2021-01-07
**Final Decision**

**Decision:**

Reject

**Comment:**

Paper proposes an approach for scene autoregressive layout generation. Four expert reviewers evaluated the paper outlining the following pros/cons of the work.

> Pros:
- Good performance across different domain [R1,R2,R3,R4]
- Formulation is general [R1,R2]
- Clever separation of different attributes [R1]
- The idea of using transformers is interesting [R4]

> Cons:
- Missing related works [R3]
- Unclear comparison with baselines that [R2]
- Lacks of  hyper-parameter tuning on the baselines [R2]
- The quantitative results do not outperform the state-of-the-art models consistently across all metric [R4]

Authors have addressed some of the concerns in the rebuttal and generally reviewers are more convinced after the rebuttal than before. The fairness of comparison to baselines remains an issue for two of the reviewers, and quality of results for one. AC acknowledges and agrees with these concerns. As such, given the large number of highly qualified submissions to ICLR and in comparison to those submissions, the paper fell slightly bellow the acceptance threshold.

That said, AC believes the approach, overall, is interesting and warrants re-submission after the appropriate revisions are implemented.